# Molecular MRI-Based Monitoring of Cancer Immunotherapy Treatment Response

**DOI:** 10.3390/ijms24043151

**Published:** 2023-02-05

**Authors:** Nikita Vladimirov, Or Perlman

**Affiliations:** 1Department of Biomedical Engineering, Tel Aviv University, Tel Aviv 6997801, Israel; 2Sagol School of Neuroscience, Tel Aviv University, Tel Aviv 6997801, Israel

**Keywords:** immunotherapy, magnetic resonance imaging (MRI), molecular imaging, treatment response, cancer, oncolytic virotherpay, artificial intelligence (AI), apoptosis, chemical exchange saturation transfer (CEST), magnetic resonance spectroscopy (MRS)

## Abstract

Immunotherapy constitutes a paradigm shift in cancer treatment. Its FDA approval for several indications has yielded improved prognosis for cases where traditional therapy has shown limited efficiency. However, many patients still fail to benefit from this treatment modality, and the exact mechanisms responsible for tumor response are unknown. Noninvasive treatment monitoring is crucial for longitudinal tumor characterization and the early detection of non-responders. While various medical imaging techniques can provide a morphological picture of the lesion and its surrounding tissue, a molecular-oriented imaging approach holds the key to unraveling biological effects that occur much earlier in the immunotherapy timeline. Magnetic resonance imaging (MRI) is a highly versatile imaging modality, where the image contrast can be tailored to emphasize a particular biophysical property of interest using advanced engineering of the imaging pipeline. In this review, recent advances in molecular-MRI based cancer immunotherapy monitoring are described. Next, the presentation of the underlying physics, computational, and biological features are complemented by a critical analysis of the results obtained in preclinical and clinical studies. Finally, emerging artificial intelligence (AI)-based strategies to further distill, quantify, and interpret the image-based molecular MRI information are discussed in terms of perspectives for the future.

## 1. Introduction

In recent decades, survival rates for several cancer types have improved through massive investments in research, earlier and wider population screening, and the discovery of new therapeutics [1]. However, cancer remains a devastating disease that still constitutes the first or second leading cause of death in most countries worldwide [2]. While surgical tumor resection is the oldest and most intuitive line of treatment, it can be a traumatic and stressful event that has undesirable effects on a patient’s quality of life. Surgical intervention can damage the surrounding healthy tissue and often leaves residual tumor cells behind [3,4,5]. As a result, further treatment modalities (particularly radiotherapy and chemotherapy) are often administered instead, before, or following the surgery [6,7,8].

Radiotherapy generates free radicals and can cause direct DNA damage aimed at depriving cells of their ability to proliferate and promoting cell death. One of the advantages of radiotherapy is that healthy cells can regenerate and survive radiotherapy-associated effects better than tumor cells [8]. This case-by-case tumor cell-radiation interaction depends on a variety of biological factors (such as intratumoral oxygen pressure and growth factor expression) [9] such that successful tumor response will lead to apoptosis, autophagic cell death, and necrosis [10,11]. The main challenge remains to further reduce risks to healthy cells, and overcome the inherent radioresistance of cancer types such as melanoma and glioblastoma [12].

Chemotherapy is the main treatment in advanced-stage disease (when no other approach is adequate), as pre-treatment for localized disease, or in combination with surgery or radiation [13]. Cytotoxic medicine primarily disrupts the cell cycle and induces apoptosis in fast-dividing cells [7]. Although new targeted chemotherapeutic drugs are under development [14,15,16,17], there is still a clear need to further reduce host tissue damage and mitigate chemotherapy-induced side effects (ranging from vomiting, dizziness, and depression to long-term nerve damage, kidney and cardiac pathologies, and secondary cancer) [18,19].

The recent formal regulatory approval of immunotherapy for several indications marked a major milestone in long-standing efforts to mitigate damage to healthy cells, and the struggle to improve cancer prognosis [20]. Compared to chemotherapy, clinical immunotherapy trials have reported improved outcomes and better tolerated side effects in challenging cancers [21]. Immunotherapy recruits the patient’s immune system to fight cancer via immune checkpoint blockade therapy [22], chimeric antigen receptor (CAR) T cell technology [23], neoantigens and cancer vaccines [24,25] or immunogenicity generating oncolytic viruses (OVs) [26,27,28,29].

Despite its promises, immunotherapy does not work for all patients, and the factors affecting individual patient responses are unknown [30]. Given these uncertainties, the tumor treatment response needs to be monitored longitudinally [31]. If a tumor is irresponsive, new treatment routes need to be considered as soon as possible, bearing in mind that the false classification of a tumor as non-responsive may result in unnecessary administration of additional chemotherapy or radiation. In addition to routine patient care, treatment response characterization is vital for determining the outcomes and endpoint of new clinical trials and defining appropriate trial candidates (e.g., when a progressive tumor constitutes an inclusion/exclusion criterion).

The most accurate way to assess treatment response is histological analysis of the tumor biopsy [32,33]. However, this approach has a number of drawbacks: (1) It is invasive by nature and therefore poses an inherent risk of complication [34], especially when the tumor is located in hard-to-reach regions or near a sensitive functional tissue. (2) Treatment response dynamics mandate frequent and longitudinal sampling which is difficult to achieve through biopsy [35]. (3) The tumor tissue is often spatially inhomogeneous and therefore prone to misrepresentation when a single biopsy site is analyzed [32,36,37].

These key shortcomings have contributed to the preference of noninvasive medical imaging in cancer treatment monitoring [38]. Tumor shrinkage is an intuitive response biomarker, and is the main pillar of many protocols for assessing the response of solid tumors [39,40]. However, these protocols typically rely on the subjective measurement of the tumor diameter in one or more axes, resulting in large inter-observer variability [41,42,43]. While computer algorithms for tumor volume detection or automatic diameter calculation can improve the reproducibility of morphometry-based treatment response analysis [44], it may take several weeks (or months) for morphometric changes to be detectable. Imaging of molecular properties, on the other hand, constitutes a potential window to a variety of biological processes (such as altered metabolism, cell proliferation, and death), which could serve as powerful treatment response biomarkers at a far earlier stage [45,46].

Clinically relevant molecular information can be obtained using a variety of imaging modalities, including positron emission and single photon emission computed tomographic imaging (PET and SPECT), ultrasound, optical imaging, X-ray computed tomography (CT), and magnetic resonance imaging (MRI) [37,47,48,49,50,51,52]. The latter stands out due to several characteristics: (1) It is an extremely versatile modality, where the image contrast can be programmed to emphasize a variety of biophysical properties; (2) It provides excellent soft tissue contrast, at any tissue depth; (3) It does not require the use of ionizing radiation; (4) It can detect molecular-based phenomena using contrast enhancing materials (similar to PET, SPECT, and ultrasound) but also via endogenous (injection-free) mechanisms.

The aim of this review is to provide an overview of molecular MRI-based methods for cancer immunotherapy treatment response monitoring. The underlying physics, computational, and biological aspects of molecular MRI are detailed along with a critical analysis of the results obtained in preclinical and clinical studies. In particular, this includes the management of pseudo-progressing tumors, a common imaging scenario with important clinical implications, and oncolytic virotherapy, a treatment modality that may improve tumor sensitivity to immune detection [26]. The final section discusses the state of the art in artificial intelligence (AI)-based strategies to further distill, quantify, and interpret image-based molecular MRI information, and then explores future perspectives.

## 2. Tumor Treatment Responses—Official Guidelines and Radiological Challenges

In the last four decades, several criteria have been put forward for treatment response monitoring. In 1981, the guidelines provided by the World Health Organization (WHO) mainly linked tumor shrinkage to treatment response [53]. Twenty years later, an international collaboration of health entities from the United States, Europe, and Canada presented more simplified, drilled-down criteria entitled RECIST, the response evaluation criteria in solid tumors [54], where tumor size dynamics were still a key component. While CT was the modality of choice in the original RECIST criteria, the updated RECIST 1.1 guidelines included MRI to as another useful modality of choice [40].

The relatively high mortality, lack of sufficiently effective treatment, and the biological complexity associated with brain cancer motivated McDonald et al. [55] to conceptualize specific guidelines for gliomas, where the evaluation criterion was tumor size estimation based on the contrast enhanced MRI or CT. However, this analysis failed to address several radio-pathological considerations:(1)The lack of a sufficiently specific treatment surrogate. Contrast enhancement in the brain reflects the blood–brain barrier disruption that often occurs in gliomas, but can also be caused by steroids, surgery, and ischemia [56];(2)Its limited ability to detect pseudo-progression, a radiological phenomenon where new or enlarged contrast-enhanced regions appear several months after the initiation of therapy, erroneously indicating that the tumor persists. Pseudo-progression is thought to affect 9–30% of all brain tumor patients [57] and typically originates from temporarily increased vascular permeability, generated by tissue inflammation [58];(3)Its limited ability to detect a pseudo-response, a radiological phenomenon where a decrease in contrast enhancement is seen in the absence of an actual response to therapy. This effect is commonplace in patients receiving antiangiogenic-based medicine [59].

To tackle these pitfalls, the Response Assessment in Neuro-Oncology (RANO) working group developed new criteria for high-grade [60] and low-grade [61] gliomas. The RANO criteria suggested waiting longer before defining tumor status to account for potential pseudo-progression and pseudo-response, and recommended basing the tumor analysis on a combination of contrast-enhanced MRI with T${}_{2}$-weighted or fluid-attenuated inversion recovery (FLAIR) images (see Section 3.1). The RANO-based categories are divided into complete response, partial response, stable disease, and progressive disease [39].

The emergence of immunotherapy, crowned by its first success stories, and rapidly followed by the proliferation of related clinical studies have gradually demonstrated that this treatment modality is not associated with pseudo-response, but is highly prone to pseudo-progression. Studies made it clear that immunotherapy-related pseudo-progression is driven by unique biological mechanisms (Figure 1) and is characterized by different temporal dynamics than chemo-radiotherapy [62]. Specifically, in both brain cancer and melanoma patients, tumors were reported to grow or new lesions formed after the administration of immunotherapy, such that a clear response was only visible months later [30,63]. Accordingly, the RANO group published specific criteria for the treatment response monitoring of glioma following immunotherapy [62]. The main difference between these criteria and many of the previously published RANO guidelines is the recommendation to wait three additional months before reaching a conclusive response classification decision, in the case where a radiological progression was observed in the six months following the initiation of immunotherapy.

Clearly, this change in guidelines is better suited to the inherent risk of pseudo-progression effects in immunotherapy, but for true non-responding tumors, it can entail heavy costs, during which long months are spent on an unsuitable therapeutic route (most critically in glioblastoma, where the median survival time is only 15 months [65]).

Thus, overall, pseudo-progression is a major diagnostic obstacle with immediate implications for the treatment route, and specific considerations for the assessment of the immunotherapy response. While recent treatment monitoring criteria have clearly acknowledged and accommodated pseudo-progression, current imaging protocol guidelines face problems of accuracy and only allow for a relatively *late* determination of tumor status.

## 3. MRI of Cancer Immunotherapy Treatment Response

The challenges posed by immunotherapy response kinetics, the secondary biological effects, and in particular pseudo-progression, point to the crucial need for an imaging approach that can visualize the tumor microenvironment dynamics, the therapeutic agent location, and the host tissue viability, as early as possible. This section aims to build an ability to understand, evaluate, and critically discuss molecular MRI for immunotherapy monitoring. To do so, it first outlines the basic principles of conventional MRI, the advantages of advanced (perfusion and diffusion) MRI, and its limitations. It then provides an extensive review of recent progress in molecular MRI for immunotherapy treatment response with representative examples. A general outline of this section as well as the main MRI-based strategies for treatment monitoring are shown in Figure 2.

### 3.1. Conventional MRI

The most commonly used MRI acquisition protocols are designed to highlight the spatial differences in density and nuclear magnetic relaxation times of water protons. Since the concentration of water molecules in the body can reach up to 55 M, and because different tissue types (such as the brain gray and white matter) are characterized by specific relaxation times, conventional MR images provide an excellent soft tissue contrast and a direct way to detect anatomical abnormalities and morphological changes. By choosing a particular set of acquisition parameters, in particular the echo and repetition times (TE and TR, respectively), images that are mainly affected by the longitudinal or transverse relaxation time (T${}_{1}$- and T${}_{2}$-weighted, respectively) can be produced. In the context of cancer (and particularly gliomas), three more sequences are typically used: T${}_{2}$-weighted FLAIR, which attenuates the signals stemming from the cerebrospinal fluid (CSF) [66] and T${}_{2}$*-weighted gradient echo, which allows for the detection of hemorrhages, and post-gadolinium-injection T${}_{1}$-weighted imaging.

Why is an exogenous contrast media needed? Although various tumor types may be well-detected using natural (endogenous) T${}_{1}$ or T${}_{2}$-weighted imaging, there are many others that are undetectable without contrast enhancement [67,68]. Gadolinium shortens the T${}_{1}$ relaxation time in its nearby water molecules leading to brightened image regions that reflect the accumulation of the contrast material. It thus facilitates the detection of the blood-brain barrier disruption which often occurs in gliomas, as well as the leakier blood vessels and their associated increased permeability, a common feature of solid tumors [69].

Unfortunately, post-gadolinium image enhancement is not reserved for gliomas or tumor progression, and may also appear in a variety of other pathologies [70] as well as in treatment-induced necrosis [71,72]. Many of these confounding effects present frequently after immunotherapy, causing a potential pseudo-progression effect [64,71,73]. On the other hand, up to one-third of malignant brain tumors do not enhance subsequent to contrast agent injection [74]. Given the above limitations, it is clear that standard MRI sequences are incapable of effectively and completely differentiating true from pseudo-effects at a sufficiently early stage [60]. This accounts for the efforts to develop advanced, alternative or complementary imaging techniques for treatment response monitoring.

### 3.2. Advanced (Non-Molecular) MRI

T${}_{1}$- and T${}_{2}$-weighted imaging is the mainstay of any clinical MRI examination. However, it mostly reflects the tissue and tumor morphometry. Cellularity and vascularity are two other physiological properties that undergo considerable changes in the course of cancer treatment, but on a faster timescale. To identify these properties, advanced MRI exploits their association with water diffusion and perfusion.

Diffusion weighted imaging (DWI) utilizes a set of time-dependent field gradients to explore the mobility of water in a defined direction [75,76]. By applying a series of linear algebra operations, a spatial map of the apparent diffusion coefficient (ADC) can be calculated, where bright pixels represent fast diffusion (e.g., in the CSF and in regions of edema and cystic necrosis) and dark pixels represent restricted diffusion. Given that malignant tumors are highly cellular, low ADC values are typically associated with an active high-grade disease [75,77]. Whereas the ADC metric is often used in clinical practice, other metrics for analyzing water diffusion have been suggested, such as diffusion kurtosis imaging (DKI) [78]. DKI can be calculated from standard DWI-acquired data, after additional post-processing that is not currently included in standard MRI software [78]. The potential benefits of this technique for immunotherapy monitoring were reported in several recent studies, including on dendritic cell treatment [79], and anti-PD-1 ICI monitoring [80]. A recent study of DKI in metastatic melanoma patients reported that cell death can be detected as early as 3–12 weeks after treatment initiation [81].

One of the hallmarks of cancer is an increased abnormal angiogenesis, where hyperperfusion tends to correlate with tumor aggressiveness [82]. Perfusion MRI typically initiates with a bolus injection of gadolinium, which facilitates the tracking of vascular dynamics. Two main contrast mechanisms are then exploited: T${}_{2}$*-based dynamic susceptibility contrast (DSC) or T${}_{1}$-based dynamic contrast enhancement (DCE). In DSC, a series of images is acquired enabling the calculation of the signal curve that depicts the temporal perfusion dynamics. Based on this signal, various metrics can be extracted. The relative cerebral blood volume (CBV, calculated from the area under the curve) is most often used [83]. While DSC is a useful perfusion protocol, its reliance on T${}_{2}$* may lead to ambiguity or incompatibility with cases where the lesion is near the sinuses or prone to susceptibility artifacts. In DCE, a set of post Gadolinium T${}_{1}$-weighted images is acquired, which correlates with the vasculature leakiness and permeability. The resulting wash-in, wash-out, and general flow characteristics may be informative of tumor malignancy, activeness, and treatment response [84].

Several clinical studies have documented the potential usefulness of rCBV and ADC for immunotherapy response monitoring [85,86]. Recently, a clinical study showed that it could differentiate pseudo- from true-progression in metastatic melanoma treated with immunotherapy [87]. In that study, the DCE-based classification of progression was significantly improved compared to that obtained based on lesion-volume; however, the area under the classification curve (AUC) remained less than 0.76.

While advanced MRI provides a clinically useful, additional source of information, its associated effects are still manifested relatively late in the course of treatment [88]. In a recent study by Cuccarini et al. [89], the dendritic cell immunotherapy treatment response was evaluated based on ADC and rCBV values (Figure 3). The findings indicated that the ability to distinguish true from pseudo-progression was possible two months after the first appearance of the classic pseudo-response effects. Similarly, in the study by Umemura et al. [87], pseudo-progression was distinguished from true progression using DCE MRI at a median of 1.8 weeks after starting immunotherapy.

In general, the advanced MRI methods of perfusion and diffusion enrich the set of radiological information available for assessing tumor state and treatment compliance. While the effects observed with these imaging protocols can be detected faster than morphometric tumor changes, they still manifest at a time scale of weeks or months, without direct specificity to immuno-related process.

### 3.3. Molecular MRI Monitoring

Molecular imaging is an interdisciplinary field which fuses molecular biology with medical imaging [37,71,90,91,92,93,94,95,96]. When exploited for immunotherapy, it holds dual promise:(1)Providing new insights into the mechanisms underlying the interactions between the host tissue, the tumor, and the immunotherapeutic agent.(2)Enabling treatment optimization on a patient-by-patient basis, through the early classification of an immunotherapy responsive or resistant tumor.

Classical molecular imaging can be obtained through a variety of modalities, but the most suitable types for both preclinical and clinical imaging in most body organs and for various cancer types are nuclear medicine (PET and SPECT) and MRI. None of these techniques are limited by tissue depth (unlike optical imaging) and are unaffected by the bones and the skull (unlike ultrasound). As stated earlier, the focus of this review is MRI-based immunotherapy molecular imaging, as it is highly versatile, utilizes non-ionizing-radiation, and provides excellent soft tissue contrast. Complementary information on the use of PET and SPECT for immunotherapy responses can be found in the following recently published original research papers [97,98,99,100,101,102,103], and reviews [104,105,106,107].

#### 3.3.1. MRI Probes

Although contrast enhancing materials are commonly used in the clinic, they are typically utilized in their “native configuration”; i.e., without additional modification for specific expression or targeting of a particular compound. In the molecular imaging context, these probes are further engineered and designed for direct amplification of the biological property of interest. For example, Huang et al. enwrapped an oncolytic adenovirus within a calcium and manganese carbonate biomineral shell [108]. When this complex reached the acidic tumor microenvironment, it released manganese ions, which resulted in T${}_{1}$ image enhancement that enabled virus detection. Du et al. loaded a chemotherapeutic drug in lyposomal nanoparticles that were decorated with PD-L1 antibodies (for specific cancer targeting) and labeled with a gadolinium-based contrast agent [109]. In breast and colon tumor-bearing mice, an image brightening was visible in the tumor region 24 h after intravenous injection.

The antitumor immune response is heavily dependent on the immune cell migration profile. For this reason, the image-based tracking of these cells may shed light on treatment efficacy [110]. Cho et al. developed an iron oxide-zinc oxide core-shell nanoparticle that can transfer antigens into dendritic cells while enabling their detection. They reported that the resulting image darkening was indicative of a tumor antigen specific T-cell response. In a more recent study, iron oxide nanoparticles (IONPs) were combined with dendritic cell activating liposomes [111]. Using the particles’ T${}_{2}$* shortening ability, the researchers were able to track the dendritic cells. Interestingly, the early stage (2 days post-therapy) T${}_{2}$* signals at the lymph nodes allowed for the categorization of the animals into non-responder, partial responder, and responder groups. Moreover, the late-stage (27 day post-treatment) tumor size was correlated with the 2-day MRI image darkening effects.

In another recent study by Lee et al., cell death protein ligand (PD-L1) expression was monitored using PD-L1 antibodies conjugated with lipid-coated IONPs [112]. A specific binding capacity was exhibited in a mouse model of temozolomide-resistant glioblastoma, which was clearly seen in the in vivo brain MRI.

Natural killer cells (NK) are another attractive target for MRI-probe-based tracking. In a study by Daldrup-Link et al., a genetically engineered human NK cell line was labeled with IONPs, and subsequently used for in-vivo distribution monitoring in a mammary tumor mouse model [113]. A clear signal darkening effect was obtained as early as 12 h after intravenous injection, thus highlighting the potential of this strategy for early treatment response assessment.

The above-mentioned MRI probes were designed to enable direct tracking of immunotherapy associated cells or viruses. However, probes can also be used to study the tumor microenvironment. For example, Liu et al. engineered a nanoprobe that can “sense” acidosis and hypoxia, and respond by Mn${}^{2+}$ release (manifested as T${}_{1}$-based image brightening) [114]. Since hypoxia is the main contributor to tumor microenvironment hostility and immunosuppression, these image changes enabled the prediction of treatment response to immune checkpoint inhibitors (ICI).

While the preclinical studies reported in this section have shown great promise for the monitoring and prediction of immunotherapy treatment response within only days or even hours, their longstanding limitation remains overcoming probe toxicity and ensuring adequate biocompatibility, tumor specificity, and safe clearance and biodistribution profiles, as strictly required for clinical translation.

#### 3.3.2. Magnetic Resonance Spectroscopy

Unlike conventional MRI which exploits the abundance of water molecules in the body, ${}^{1}$H Magnetic resonance spectroscopy (MRS) aims to measure the signals from other biological compounds and metabolites, at concentrations smaller by four orders of magnitude. Following an acquisition protocol that aims to suppress the water signals, the existence of a variety of compounds such as N-acetylaspartate (NAA), creatine, choline, lactate, lipids, myo-inositol, and glutamate/glutamine becomes detectable [115]. Based on the different chemical shifts of each compound, the metabolic tissue composition can be characterized. The immediate advantage of MRS compared to MRI probes is the endogenous origin of the image contrast. However, to compensate for the low metabolite concentrations, long acquisition times and a single large voxel are typically used. A multi-pixel MRS imaging variant (termed MRSI, or chemical shift imaging) may also be performed, although it still ends up with relatively large voxels.

Choline-containing compounds are one of the most interesting MRS targets, which are found in the brain myelin and cell membranes. Increased choline signals were shown to correlate with tumor proliferation and malignancy [116], and represent the rapid membrane synthesis in proliferating tumor cells. Importantly, the choline/creatine ratio was successfully exploited to differentiate the true progression from treatment induced effects in several clinical brain cancer studies [117].

The fatty acids associated with tumor cell membranes are yet another important treatment biomarker which correlates with apoptosis [118]. Limmatainen et al. reported the concomitant changes occurring in choline and lipid MRS signals in an apoptotic rat glioma model [119]. Interestingly, on day 8 of ganciclovir treatment, the lipid (apoptotic representative) signal reached its peak, while the choline signal, which represents tumor malignancy, started abruptly decreasing. A later carcinoma rodent study by Hemminki et al. showed that a combination of these signals was useful for OV treatment monitoring, and could detect true responders in a matter of days following treatment initiation [120].

#### 3.3.3. ${}^{19}$F MRI

Although ${}^{1}$H MRS is a powerful tool for studying the chemical composition of tissues, it lacks specificity for labeling and tracking specific cells. ${}^{19}$F MRI, on the other hand, utilizes exogenously administered fluorine to provide metabolic cell information, which is considered unambiguous (given the scarcity of endogenous ${}^{19}$F in the body) [121]. Recently, Croci et al. successfully used perfluorocarbon-containing nanoparticles and ${}^{19}$F MRI to noninvasively track tumor associated microglia and macrophages over time in response to radiotherapy [122].

Weibel et al. imaged intratumor inflammation using perfluorocarbon-labeled immune cells. They applied this methodology to a variety of animal tumor models undergoing OV-treatment, and demonstrated its potential for early stage treatment monitoring [91].

Similar to the IONP reports presented in Section 3.3.1, a rodent tumor study by Ahrens et al. showed that ${}^{19}$F MRI can be used to monitor dendritic cell migration to the lymph node [123]. Dubois et al. imaged perfluorocarbon labeled CAR T-cells in a mouse model of leukemia at a clinical 3T field strength [124]. The detection was enabled as early as 1 day post-injection, and the perfluorocarbon did not alter the cytotoxic profile.

Overall, ${}^{19}$F MRI provides very high specificity in detecting molecular phenomena, but is hampered by the need for specialized radiofrequency coils that are not included in standard MR scanners.

#### 3.3.4. Hyperpolarized Carbon-13 MRI

Similar to ${}^{19}$F imaging, ${}^{13}$C MRI exploits nuclear species that are rare in living organisms. To benefit from the inherent specificity associated with the exogenous administration of these compounds to the body but still be able to sufficiently amplify their signals, a process called dynamic nuclear polarization needs to be used which requires dedicated instruments, very low temperatures, and strong magnetic fields. Eventually, a substrate of interest becomes MR-visible and its metabolism (after intravenous injection) can be characterized [125].

Thus, traditional ${}^{1}$H MRS provides a snapshot of the molecular state, whereas hyperpolarized ${}^{13}$C MRI provides a dynamic and rapid window to investigate metabolic changes. There are a number of endogenous metabolites that can be labeled with ${}^{13}$C, of which pyruvate has been the most extensively studied. In the context of cancer, tumors consume more pyruvate than healthy cells, as described by the Warburg effect [126]. As a result, the conversion of pyruvate into lactate is correlated with tumor malignancy [127]. Miloushev et al. studied the ${}^{13}$C pyruvate to lactate dynamics in a small patient cohort composed of oligodendroglioma, recurrent glioblastoma, melanoma, and metastatic ovarian carcinoma patients. They observed stronger lactate signals in untreated as compared to treated tumors, as well as in recurrent tumors [128]. In a more recent study, ${}^{13}$C-based pyruvate to lactate conversion rate maps were reconstructed, and enabled the detection of the early metabolic response to ICI in a prostate cancer patient [129].

In addition to the direct imaging of metabolism, ${}^{13}$C MRI has also been used to investigate the tumor microenvironment. Gallagher et al. labeled bicarbonate with hyperpolarized ${}^{13}$C to obtain quantitative pH images in a subcutaneous mouse tumor model [130]. A year later, the same group injected hyperpolarized ${}^{13}$C-labeled fumarate intravenously into lymphoma-bearing mice, and were able to image its conversion to malate. The rate of this process served a biomarker for cell necrosis and early treatment response [131].

The practical implementation of ${}^{13}$C MRI requires ad hoc preparation of the injected solutions, the purchase of a polarizer and X-nuclei MRI coils, and eventually an injection and imaging under strict time constraints (since the signal attenuates within minutes). However, in light of the promising quantitative, metabolic, and treatment specific molecular information it has already provided, and if further confirmed in large cohort clinical studies, the associated financial and technical burden will be largely compensated for.

#### 3.3.5. Chemical Exchange Saturation Transfer (CEST) MRI

CEST is a molecular imaging technique that enables the imaging of metabolites, mobile proteins and peptides [132,133,134,135]. Unlike MRS, CEST enables relatively high spatial resolution imaging; e.g., a pixel size of several hundreds of microns or 1–2 mm in preclinical and clinical studies, respectively, while retaining reasonable scan times. In a typical CEST acquisition protocol, a series of saturation pulses are applied at a variety of chemical shifts around the water resonance frequency. When the saturation pulse frequency matches the resonance frequency of the exchangeable protons of interest, their net magnetization is decreased at which point these protons undergo a chemical exchange with the water protons, yielding a decreased water-pool signal. By applying sufficiently long (on the order of seconds) saturation pulses, or saturation pulse trains, a dramatic signal amplification can be achieved, allowing for the imaging and detection of millimolar compound concentrations [136]. Some of the most useful endogenous compounds that contain exchangeable protons include the amide (–NH), amine (–NH${}_{2}$), and hydroxy groups (–OH) [49]. The CEST signal depends on the volume fraction of the labile protons (which is proportional to the compound or metabolite concentration), the exchange rate between the solute and water protons (which is typically correlated with the temperature and pH) [137,138], water relaxivity, and the acquisition parameters [139]. Since the CEST signal originates from molecular compounds and is influenced by the microenvironment, it can inform on early-occurring treatment-related effects.

##### Treatment Monitoring Using Endogenous CEST Contrast

The most widely used CEST target consists of the amide protons in mobile cellular proteins and peptides [140]. In a seminal study by Zhou et al. [96], the CEST-weighted amide signal (at 3.5 ppm downfield of the water resonance frequency) was able to distinguish tumor recurrence from radiation necrosis. In rats, cell death was manifested as CEST-image darkening, due to the absence of mobile cytosolic proteins and peptides. In active glioma on the other hand, image brightening was obtained due to the increased cytosolic protein and peptide content. Importantly, the T${}_{1}$-weighted post-gadolinium imaging of the same animals was not able to distinguish between the two groups. Based on the same contrast mechanism, Sagiyama et al. [141] were able to detect an early therapeutic response to chemotherapy in glioblastoma-bearing mice. The amide CEST signal decrease, which was obtained 4 days post-chemotherapy, was concordant with the reduced levels of cell proliferation, as seen in Ki67-based histology, and preceded any changes in tumor volume. The amide CEST signal has been further investigated in a variety of clinical studies that could differentiate between pseudo-progression and true tumor progression [142,143].

In addition to its sensitivity to mobile protein concentrations, the dependence of the amide CEST signal on pH (via the proton exchange rate) [144,145] can serve as another important treatment biomarker, given that increased intracellular (and decreased extracellular) pH is the hallmark of cancer [146,147]. Additional CEST targets, such as the amine protons at 3 ppm downfield of water have a linear signal relationship with pH in the physiological range [148]. Finally, based on the combination of amine and amide signal ratios, an absolute pH mapping, independent of compound concentration can be performed [149].

Recently, Cho et al. presented preliminary data showing the ability of amine-weighted CEST for the detection of an immunotherapy response [150]. In a rodent study, the tumor CEST signal at 3 ppm was elevated at baseline, and returned to normal (contralateral-tissue-like) values after a combination of dendritic cell vaccination and anti-PD1 treatment (Figure 4A). The authors also presented single-patient data that were consistent with the pre-clinical findings, where a recurrent glioblastoma patient was successfully treated with PD-L1. The corresponding amine-CEST images were in visual agreement with the post-contrast T${}_{1}$-weighted findings (Figure 4B).

In a different study, the applicability of endogenous amide-based CEST MRI for treatment monitoring was examined in a glioblastoma mouse model of oncolytic virotherapy [151]. However, the standard CEST analysis metrics (based on the signal asymmetry around the water resonance frequency and Lorentzian model fitting) failed to discriminate between the apoptotic treatment responsive core and the tumor periphery (which was unaffected by the virus). This was apparently caused by the multitude of confounding signals stemming from the altered water relaxivity (associated with edema), the semisolid macromolecule magnetization transfer dynamics, and the acquisition parameters used. To overcome these obstacles, an advanced artificial-intelligence-boosted acquisition and reconstruction technique was developed (see Section 4), which resulted in the spatial quantification of the amide proton exchange rate and volume fraction across the animal brain. Both of these parameters served as useful biomarkers for apoptosis detection (Figure 5), which is characterized by protein synthesis inhibition and decreased intracellular pH [152,153].

##### Treatment Monitoring Using Exogenous CEST Contrast

Given the complexity of the in-vivo molecular environment, the existence of multiple proton pools, and the simultaneous changes in several magnetic tissue properties during cancer treatment, it is highly desirable to further increase the signal-to-noise ratio, sensitivity and specificity of CEST imaging using exogenously injected materials. In recent decades, a variety of CEST agents have been developed, based on paramagnetic [154,155] and diamagnetic [156] materials, liposomes [157], iodine [158,159], and glucose [160,161,162,163].

One of the main hurdles to MRI probe development is the need to provide sufficient and extensive proof of biocompatibility, followed by exhaustive examinations and trials before a regulatory approval for clinical use may be considered. In a clever re-purposing approach, it was found that previously established and FDA approved iodine-based CT contrast agents may also serve as CEST contrast media. These compounds contain a few amide groups that resonate at different frequencies [164,165]. By calculating the ratio of the associated CEST signals, the extracellular pH can be derived and mapped across the tumor [166]. Acidosis is not only a hallmark of malignancy but also promotes oncogenetic metabolism. Proton pump inhibitors (PPI) are molecules aimed to inhibit the extracellular tissue acidification by creating an alkaline pH that may improve the efficiency of immunotherapy [167]. A recent preclinical study by Irrera et al. reported that CEST MRI with the CT contrast agent iopamidol was able to successfully monitor the pH changes associated with PPI administration [168].

Reporter gene imaging allows for the monitoring of transgene expression, and hence constitutes a powerful technique for exploring the molecular processes involved in immunotherapy. In a seminal work in the field, Gilad et al. designed an artificial CEST reporter gene based on the biodegradable lysin rich-protein (LRP) [169]. It creates a detectable contrast at the amide proton frequency, which enables the milli-molar-level detection of cells in tumors. In a more recent study, Farrar et al. engineered the LRP reporter gene into a herpes simplex-derived oncolytic virus [170]. The virus was then inoculated into glioma bearing rats in-vivo, and CEST-weighted images were acquired at baseline and in the acute phase (Figure 6). A significant increase in CEST contrast was observed within the virally-infected tumors, while retaining viral replication and therapeutic efficacy. Although the LRP reporter gene has been established in a variety of in-vivo studies [50,171], and further redesigned for increased stability and sensitivity [172], it is limited by the confounding endogenous tissue amide proton signals, which overlap with the LRP effect (at around 3.5 ppm). Bar-Shir et al. recently designed a CEST reporter gene with labile exchangeable protons that resonate at a large chemical shift (5.8 ppm) [173]. They employed this reporter to track dendritic cell cancer vaccine in a mouse brain tumor model and reported it could detect the presence of fewer than 10,000 cells.

## 4. Artificial Intelligence (AI) in Immunotherapy Treatment Monitoring

Despite the major developments in the field of molecular MRI, and their direct applications to immunotherapy response monitoring, a few key limitations remain unresolved that hinder the translation of these methods into widespread clinical use:(1)The acquisition time may be relatively long, especially for inherently low signal-to-noise ratio (SNR) methods;(2)The images are mostly qualitative and depend on the particular set of acquisition parameter used;(3)The clinical data interpretation (and final tissue state characterization) is observer-dependent.

It is hard to ignore the major leap in AI performance, whose direct implications affect almost all fields of science. Medical imaging in particular has benefited from this progress, in that many modern AI frameworks were originally designed to mimic human visual perception [174,175]. Luckily, these opportunities were not overlooked by molecular imaging scientists, who started meeting the challenges mentioned above by incorporating and expanding AI strategies to boost the performance, accuracy, and clinical compatibility of molecular MRI. AI in medical imaging encompasses an extremely broad field, based on advanced mathematical and computational foundations, with a large volume of treatment response-related publications [176,177,178]. This section provides the reader with some concrete examples of the potential benefits of “injecting” AI throughout the molecular imaging pipeline (Figure 7), while highlighting the works related to immunotherapy. In this brief guided tour, we will travel backwards in the imaging process, starting from the last classification step, all the way back to the acquisition protocol design.

While only occurring at the end of the imaging process, the use of AI is most intuitive in the context of final tissue classification, which is aimed to derive an automatic and inter-observer variability-free assessment of tissue response. A variety of methods have been developed for this task, which typically employs conventional MRI. For example, Chang et al. proposed a method for automatic RANO glioma assessment [44]. It consists of two main parts: a 3D U-net convolutional neural network that segments FLAIR and T${}_{1}$ MR images, and an additional algorithm that calculates the required RANO measurements from the segmented masks. Cho et al. implemented a similar strategy but employed automatic RANO on volumetric information [179]. Vollmuth et al. expanded the input image modalities to include diffusion information [180]. Although still using RANO criteria and conventional imaging, Rundo et al. proposed a semi-automatic approach where the radiologist manually selects the region of interest followed by automatic morphometric analysis and deep *immunotherapy* response classification [181]. In a more recent study, Guo et al. [182] implemented deep learning techniques on a combination of conventional MR and amide CEST-weighted images to separate true tumor progression from radiation effects. In this study of glioma patients, the addition of molecular images as input improved diagnostic performance.

Notably, many tissue state classification techniques rely on a prior delineation of the tumor margins. Since this can be achieved using conventional MR images, it is beyond the scope of this review (see comprehensive reviews in [183,184,185]).

One of the critical limitations of medical imaging is the contrast-weighted form of commonly acquired data. While exact biophysical parameter maps can be extracted using repeated acquisition at steady state conditions and known analytical solutions (as in classical water T${}_{1}$ and T${}_{2}$ mapping), this approach translates into long acquisition times, which are impractical for clinical use. This issue is further exacerbated in molecular imaging, where the low SNR demands even longer acquisition times, and where the complexity of the biological scenario (which involves a number of proton pools, especially in the brain) may severely undermine quantification accuracy and further increase its computation time.

To address these quantification challenges, a variety of AI-based methods have recently been developed. Gurbani et al. proposed a physics-driven deep learning model to accelerate the spectral fitting of MRSI [186]. In a study of glioblastoma patients, whole-brain quantification took less than 1 min. Magnetic resonance fingerprinting (MRF) is a relatively new paradigm in quantitative MRI [187]. Initially proposed for conventional MRI, it allows the simultaneous extraction of water T${}_{1}$ and T${}_{2}$ in sub-minute scan time. MRF is built on pseudo-random and rapid acquisition protocols that generate a large set of heavily under-sampled and noisy raw images. However, the temporal dynamics hidden in each pixel’s trajectory contain a unique signal pattern (fingerprint), which represents its underlying biophysical tissue properties. By exploiting the well-studied Bloch equations, an extensive pre-experiment simulation can generate a large dictionary of expected signal trajectories for a variety of tissue parameter combinations. By comparing the simulated dictionary entries to the acquired experimental data, the inverse quantification problem can be solved, and the underlying magnetic properties determined. In the molecular imaging realm, MRF was successfully expanded for MRS [188] and CEST quantification [189,190,191,192]. To account for the exponential growth in dictionary size, which translates into very long molecular parameter matching, a variety of deep-learning architectures have been suggested. These works enable rapid image acquisition (e.g., <5 min) and whole brain molecular parameter quantification in a matter of seconds [151,193,194,195,196,197]. Importantly, in a virotherapy treatment monitoring animal study, a comparison of conventional CEST-MRF to deep-learning-based CEST-MRF pointed to the clear superiority of the latter in terms of quantification accuracy and speed [151].

Moving further back in the molecular MRI pipeline leads to the interface between the k-space (the spatial frequency domain) and the raw image data reconstruction. While this field has been extensively explored [198,199], and has resulted in exciting AI-associated breakthroughs [200], it is nonspecific for molecular imaging.

Finally, in recent years, researchers have begun to explore the feasibility of harnessing AI-based strategies at the earliest possible intervention point, namely, for acquisition protocol design. Zhu et al. represented the analytical solution of the Bloch equations as a computational graph forming a supervised learning architecture that can update and optimize the protocol parameters (e.g., the flip angle (FA) and TR) to generate very rapid acquisition schemes that can quantify water T${}_{1}$ and T${}_{2}$ while overcoming field inhomogeneities [201,202,203].

Loktyushin et al. proposed a supervised learning framework to discover MRI sequences for different tasks, including k-space trajectory and flip angle optimization, specific absorption rate (SAR) mitigation, and quantitative T${}_{1}$ mapping [204]. Perlman et al. developed an end-to-end framework for generating rapid (less than 1 min) CEST acquisition protocols while simultaneously training a reconstruction network that extracts quantitative molecular maps from the raw data [205]. Kang et al. developed a learning-based approach for the optimization of semisolid magnetization transfer MRF protocols [206] and demonstrated its applicability on human subjects.

While machine-learning based methods for acquisition protocol design are typically based on well-established magnetic signal propagation models, two very recent works have shifted the optimization stage from the pre-experiment offline, simulative environment to an actual physical scanner. Beracha et al. developed an adaptive MRS acquisition optimizer [207]. By applying repeated acquisition while using the measured signal and information-theory optimal bounds to update the next acquisition parameters in real time, the authors successfully extracted the relaxation time of in vivo metabolites, while achieving a 2.5-fold acceleration. Glang et al. developed a model-free target driven framework, where the scanner’s “actions” are directly linked to the image target [208]; for example, the concentration of the metabolite of interest. By allowing the scanner to freely optimize its acquisition profile until the output data converges to the desired target, the authors generated new quantitative CEST protocols within a 3 h acquisition-based optimization. Although extremely innovative, additional research needs to be conducted to overcome the need for very long acquisition-based optimization and the availability of a ground-truth reference, which are challenging to obtain in-vivo.

## 5. Conclusions and Outlook

In this review, today’s key methods for molecular MRI-based immunotherapy treatment monitoring were presented, compared, and discussed. These techniques enable the tracking of cells and viruses, the derivation of tissue pH, and imaging of hypoxia, protein expression, cell death, and metabolite concentration, which ultimately translates into *early* detection of treatment response, the core advantage of molecular MRI.

The main hurdles include the inherently low SNR of the contrast mechanisms (as in MRS), the need for expensive and non-conventional equipment (such as X-nuclei coils and a polarizer), and high field MRI (which is useful for MRS and CEST imaging). Since most of these emerging approaches have been presented in animal studies, large cohort clinical studies are needed and regulatory approval obtained prior to widespread adoption.

Although each imaging route (conventional, advanced, and molecular MRI) provides its own biologically interesting layer of information, the future is likely to lie in a multi-modal combination of all these rich data into a single imaging and classification pipeline. However, the more protocols are added to the clinical routine, the longer the scan time per patient becomes, which results in increased cost and patient discomfort. Given the massive ongoing research efforts to accelerate MRI acquisition in general and molecular MRI in particular (see Section 4), we expect that the benefits of incorporating molecular MRI into the cancer treatment monitoring pipeline will outweigh its (current) shortcomings.

As discussed in the previous section, AI-based strategies can contribute to accelerating the acquisition and reconstruction of molecular MRI, improve its quantification performance, denoise its output, and automate protocol design and ultimate tissue classification. Given the success of deep learning in non-image related fields, new discoveries will doubtless be fused with molecular MRI in the future, such as text processed from electronic health records [209] and blood test data [210], to potentially uncover hidden biological relations and achieve a better characterization of tissue state. However, since all these disruptive technologies will need to be acknowledged, accepted and routinely practiced by physicians, it is crucial to ensure that they become more transparent [211] and explainable [212].

In terms of repeatability and reproducibility, expert-based consensus guidelines are also a must (though some have recently started to emerge) for reporting standards [213], the use of optimized and shared acquisition protocols [214,215], and the derivation of “good-method-development-practices” [216] for molecular and quantitative MRI method development.

As shown in this review, a variety of molecular MRI-based methods have been developed for studying the underlying mechanisms responsible for cancer immunotherapy responses, and providing early in vivo detection of tumor and host tissue state. Given the promising preliminary results and the continuous improvement in imaging accuracy, speed, and specificity, this imaging modality is expected to potentially become a valuable tool for routine clinical management of immunotherapy.

## Figures and Tables

**Figure 1 ijms-24-03151-f001:**
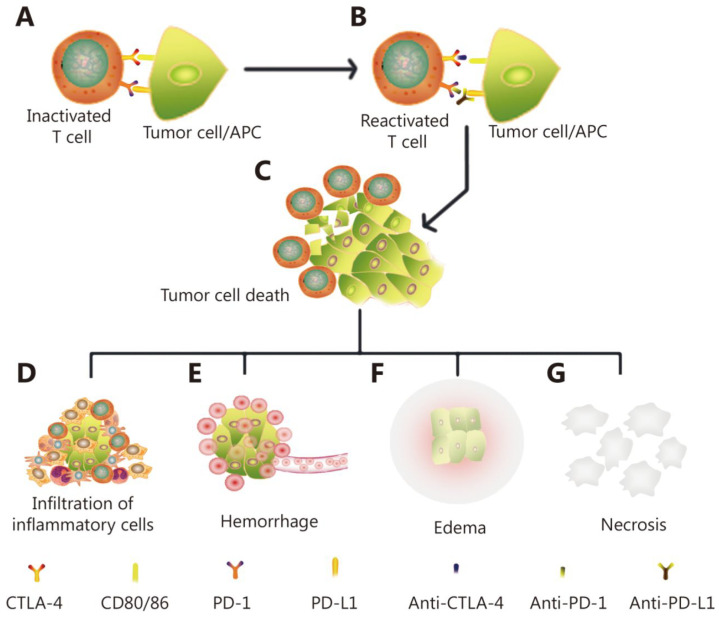
A mechanistic illustration of immunotherapy induced pseudo-progression, where the immune system response to treatment falsely appears in medical images as progressive cancer. Initially, inactivated T cells (**A**) are reactivated (**B**) subsequent to the administration of immune checkpoint inhibitors. This yields tumor cell death (**C**) and releases antigens which attract additional infiltrating inflammatory cells (**D**). The tumor shrinkage then causes vascular tears and hemorrhage (**E**), which is expressed as lesion edema (**F**). The combination of necrotic tumor cells (**G**), inflammation (**D**), hemorrhage (**E**), and edema (**F**) is ultimately (and falsely) expressed as tumor progression when examined using conventional clinical MRI. Reproduced from [64]. Copyright ©2019, Cancer Biology & Medicine (CC BY 4.0 license).

**Figure 2 ijms-24-03151-f002:**
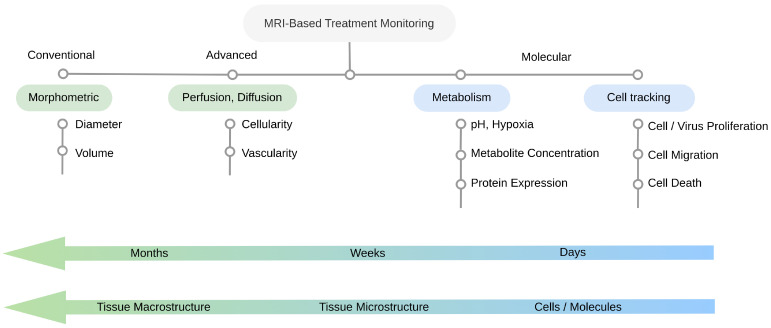
MRI-based approaches for immunotherapy response monitoring. Conventional MRI may detect morphometric changes, which appear at a relatively late stage (months after treatment initiation). Advanced (perfusion and diffusion) MRI enables the detection of cellularity and vascularity-related changes, which appear after weeks or months, without direct specificity to the immune processes. Molecular MRI enables specific cell or virus tracking or the imaging of metabolism, thus providing the earliest time window for immunotherapy response characterization.

**Figure 3 ijms-24-03151-f003:**
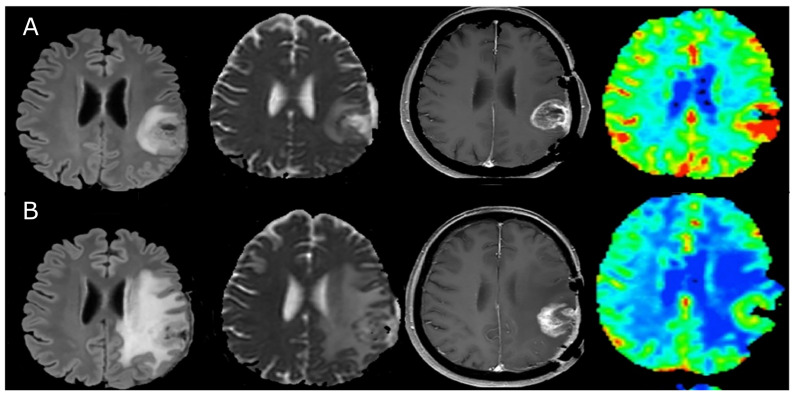
A single subject example of the radiological manifestation of pseudo-progression during immunotherapy, as seen in conventional and advanced MRI. Left to right: FLAIR, ADC, post-gadolinium T${}_{1}$, and CBV. In the first MRI scan (**A**) taken after surgery and radio-chemotherapy before immunotherapy initiation, the increased T${}_{1}$ enhancement and CBV are apparent. An edema is seen in the FLAIR image and decreased ADC, which suggests increased cellularity. In the MRI image set taken four months after immunotherapy (**B**), the tumor volume appears to have increased in the T${}_{1}$ image, but the CBV has decreased and the ADC values are higher than in (**A**), indicating decreased vascularity and less restriction, respectively. These changes suggest that the apparent increase in tumor volume is actually misleading and is driven by pseudo-progression. Adapted with permission from ref. [89], Cuccarini et al. J. Clin. Med. 2019;8;2007 (CC-BY 4.0 license).

**Figure 4 ijms-24-03151-f004:**
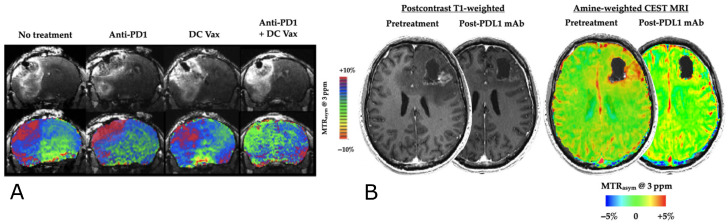
Immunotherapy treatment response monitoring using amine-weighted CEST imaging. (**A**) Postcontrast T${}_{1}$-weighted images (top) and amine-weighted CEST images (bottom) obtained during combination immunotherapies with dendritic cell vaccination and anti-PD-1 treatment. A response to the treatment is manifested as a decreased CEST signal (green compared to red in the left-hand side of the brain); (**B**) post-contrast T${}_{1}$ (left) and amine-weighted CEST (right) images from a recurrent glioblastoma patient treated with PD-L1 demonstrating a successful response characterized by homogenization of the CEST Amine contrast. Reproduced with permission from Cho et al. NMR in Biomedicine 2022;e4785. doi:10.1002/nbm.4785 [150]. Copyright ©2022 John Wiley & Sons Ltd.

**Figure 5 ijms-24-03151-f005:**
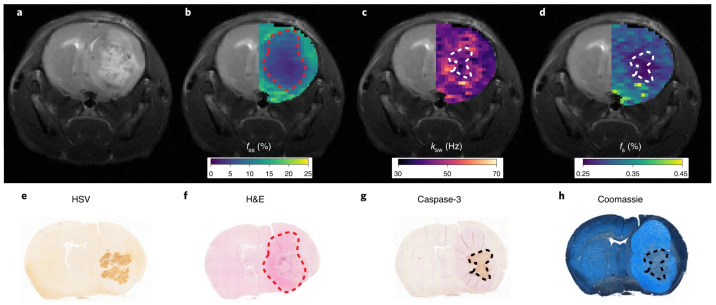
Quantitative CEST imaging of OV-induced apoptosis in a glioblastoma-bearing mouse, 72 h following virus inoculation. (**a**) conventional T${}_{2}$-weighted image; (**b**) semisolid macromolecule proton volume fraction (f${}_{ss}$) map, where a decreased volume fraction represents tumor-related edema and a change in the lipid composition of tumor tissue relative to normal brain tissue; (**c**,**d**) amide proton exchange rate (k${}_{sw}$, **c**) and volume fraction (f${}_{s}$, **d**) maps. Regions of decreased intracellular pH and mobile protein concentration, respectively, are indicative of apoptosis; (**e**–**h**) histology and immunohistochemistry images validate the MR findings with cleaved caspase-3 positive tumor regions and decreased Coomassie blue protein staining, indicative of apoptosis. Reprinted with permission from Perlman et al. Nat Biomed Eng. 6, 648-657 (2022) [151]. Copyright ©2021, The authors, under exclusive licence to Springer Nature Limited.

**Figure 6 ijms-24-03151-f006:**
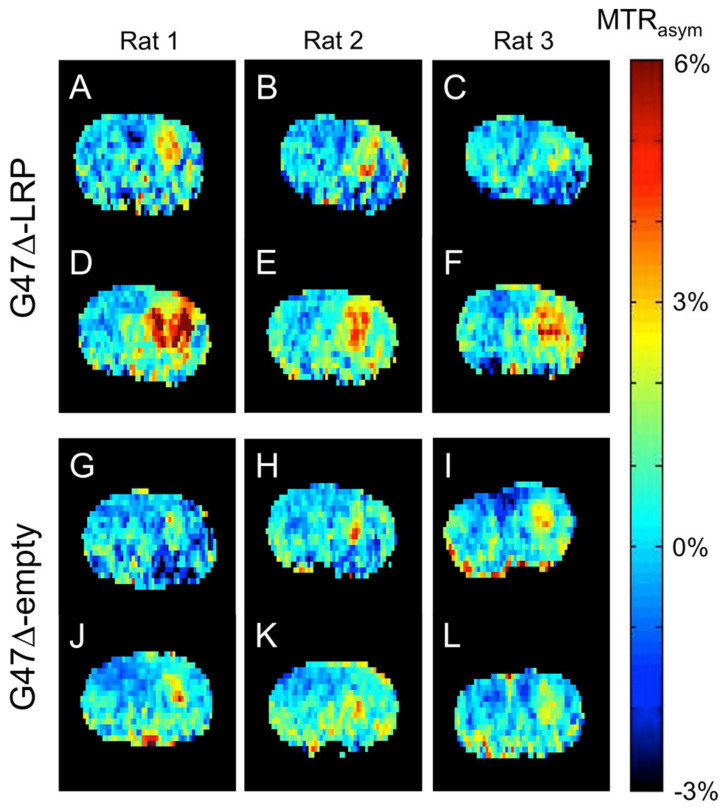
Oncolytic virotherapy monitoring using a CEST reporter gene. CEST-weighted images were acquired from three rats before (**A**–**C**,**G**–**I**) and 8–10 h after (**D**–**F**,**J**–**L**) the inoculation of an LRP expressing (top) or empty (bottom) G47Δ virus. An increased CEST signal at the tumor was obtained for the LRP OV expressing group (**D**–**F**) but not in the control group (**J**–**L**). Reprinted with permission from Farrar et al. Radiology 275(3), pp. 746–754 (2015) [170]. Copyright ©2015 Radiological Society of North America (RSNA).

**Figure 7 ijms-24-03151-f007:**
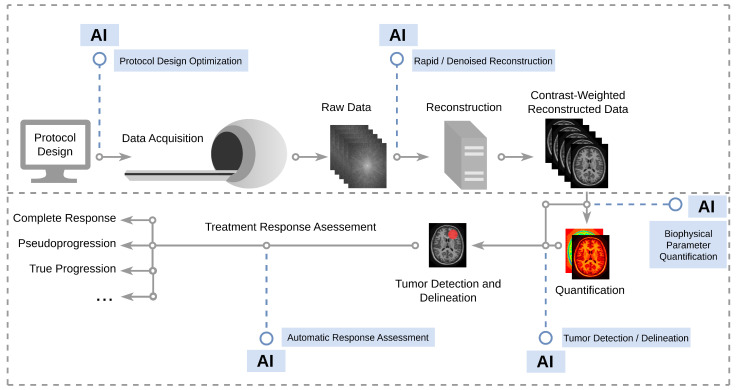
AI-based interventions along the imaging pipeline, allowing automatic and optimized protocol design, faster and denoised image reconstruction, rapid quantification of biophysical parameters, automatic tumor delineation, and automatic classification of tumor responsiveness.

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
