# Peer review of "Molecular MRI-Based Monitoring of Cancer Immunotherapy Treatment Response"

_ijms, 2023, doi:10.3390/ijms24043151_

Round 1
Reviewer 1 Report
Thank you for the opportunity to review the article entitled:
Molecular MRI-Based Monitoring of Cancer Immunotherapy Treatment Response
General comments:
The article is well-written and highly interesting. As a comprehensive literature review, the manuscript deserves attention.
Specific remarks:
Abstract and keywords: The Abstract is concise and informative; the keywords are appropriate.
Introduction: The Introduction is very interesting and includes all the important data this section should provide. The number and choice of references are appropriate. One remark: I would kindly advise mentioning one very important limitation of the surgical intervention. Although the Authors were kind to say some side effects already, the matter of quality of life (QoL) of the patient has not been raised. The surgical intervention affects the tumour/tissue and the patient. First, the intervention is highly stressful and possibly limits the patient’s QoL (i.e. organ removal); second: it is not always possible due to a low general health condition. It is essential when the radical treatment is considered (i.e. oesophagal cancer - often, the element of the aggressive protocol cannot be introduced to the patient as per his low general health condition).
Tumour treatment responses - official guidelines and radiological challenges: It seems important to mention those elements, and I sincerely appreciate this section.
MRI of cancer immunotherapy treatment response: One, strictly professional remark – technician provides the MRI images due to the competencies of the operator.
“Imaging the nuclear magnetic relaxation times is the bread and butter of any clinical MRI exam” - seems overly general (predominantly: “nuclear” plus the very common phrase. I would kindly advise using simple, scientific language instead of colloquial speech).
“A recent study of DKI in metastatic melanoma patients reported that cell death may be 213 detected as early as 3-12 weeks” – after establishing the diagnosis? Please, specify.
Section 3.2 provides several data based on the original reports. I would kindly suggest data ordering. It seems highly valuable to make sure that one case/diagnosis (i.e. melanoma malignant) undergoes discussion in one, concise paragraph. One utility – one paragraph.
I would kindly suggest avoiding expressions of option choice such as “x/y” and providing one of these options.
Artificial intelligence (AI) in immunotherapy treatment monitoring: It is highly appreciated that the Authors decided to provide this element as AI is currently one of the “hottest” topics in medical literature.
Conclusions and outlook: Although I agree with the conclusion, one element is missing. Please, consider adding a paragraph discussing the influence of the method on the patients’ management and the patients themselves. Could the method be used as an independent method? Does it answer all the clinical questions and concerns? What is the possibility that the tool presented by the Authors shortens the clinical pathway and influences overall patient management? The more studies we need to perform, the more it weighs on the patient. What are the main advantages and the most critical limitations of the tool? Please, indicate the practical dimension of the research.
Finally, the Material and Method should be provided. How did the Authors choose the literature the article is based on? What were the inclusion/exclusion criteria of sources? Usually, the appropriate flow chart or scheme is provided in such reports.
Reviewer 2 Report
Vladimirov and Perlman present an interesting and timely review article about the potential of applying advanced MRI imaging techniques and AI technology in monitoring patient response to cancer immunotherapy. The presentation is comprehensive and well-written. It would be interesting to include a discussion of the practicality of implementing these techniques in terms of the cost and availability of contrast agents. Also, the need for specialized personnel and equipment could be a barrier that I suggest to be discussed.
